# Electrochemical DNA Biosensor Based on Mercaptopropionic Acid-Capped ZnS Quantum Dots for Determination of the Gender of Arowana Fish

**DOI:** 10.3390/bios12080650

**Published:** 2022-08-17

**Authors:** Eka Safitri, Lee Yook Heng, Musa Ahmad, Ling Ling Tan, Nazaruddin Nazaruddin, Khairi Suhud, Chew Poh Chiang, Muhammad Iqhrammullah

**Affiliations:** 1Department of Chemistry, Faculty of Mathematics and Natural Sciences, Universitas Syiah Kuala, Darussalam, Banda Aceh 23111, Indonesia; 2Department of Chemical Sciences, Faculty of Science and Technology, Universiti Kebangsaan Malaysia, Bangi 43600, Selangor Darul Ehsan, Malaysia; 3Southeast Asia Disaster Prevention Research Initiative (SEADPRI-UKM), Institute for Environment and Development (LESTARI), Universiti Kebangsaan Malaysia, Bangi 43600, Selangor Darul Ehsan, Malaysia; 4Chemical Technology Program, Faculty of Science and Technology, University Sains Islam Malaysia (USIM), Nilai 91800, Negeri Sembilan, Malaysia; 5Freshwater Fisheries Research Division, Fisheries Research Institute Glami Lemi, Jelebu 71650, Negeri Sembilan, Malaysia; 6Department of Life Sciences and Chemistry, Jacobs University Bremen, 28759 Bremen, Germany

**Keywords:** electrochemistry, gold nanoparticles, screen printed electrode, DNA biosensor, ZnS QDs

## Abstract

A new electrochemical DNA biosensor based on mercaptopropionic acid (MPA)-capped ZnS quantum dots (MPA-ZnS QDs) immobilization matrix for covalent binding with 20-base aminated oligonucleotide has been successfully developed. Prior to the modification, screen-printed carbon paste electrode (SPE) was self-assembled with multilayer gold nanoparticles (AuNPs) and cysteamine (Cys). The inclusion of MPA-ZnS QDs semiconducting material in modified electrodes has enhanced the electron transfer between the SPE transducer and DNA leading to improved bioanalytical assay of target biomolecules. Electrochemical studies performed by cyclic voltammetry (CV) and differential pulsed voltammetry (DPV) demonstrated that the MPA-ZnS QDs modified AuNPs electrode was able to produce a lower charge transfer resistance response and hence higher electrical current response. Under optimal conditions, the immobilized synthetic DNA probe exhibited high selectivity towards synthetic target DNA. Based on the DPV response of the reduction of anthraquinone monosulphonic acid (AQMS) redox probe, the MPA-ZnS QDs-based electrochemical DNA biosensor responded to target DNA concentration from 1 × 10^−9^ μM to 1 × 10^−3^ μM with a sensitivity 1.2884 ± 0.12 µA, linear correlation coefficient (R^2^) of 0.9848 and limit of detection (LOD) of 1 × 10^−11^ μM target DNA. The DNA biosensor exhibited satisfactory reproducibility with an average relative standard deviation (RSD) of 7.4%. The proposed electrochemical transducer substrate has been employed to immobilize the aminated Arowana fish (*Scleropages formosus*) DNA probe. The DNA biosensor showed linearity to target DNA from 1 × 10^−^^11^ to 1 × 10^−^^6^ µM (R^2^ = 0.9785) with sensitivity 1.1251 ± 0.243 µA and LOD of 1 × 10^−^^11^ µM. The biosensor has been successfully used to determine the gender of Arowana fish without incorporating toxic raw materials previously employed in the hazardous processing conditions of polypyrrole chemical conducting polymer, whereby the cleaning step becomes difficult with thicker films due to high levels of toxic residues from the decrease in polymerization efficacy as films grew.

## 1. Introduction

Electrochemical detection provides high sensitivity, fast and simple operation conditions for the detection of various types of target analyte of interest. The electroanalytical method is independent from the color or turbidity of the solution, compatible with micro fabrication and has a facile assay method [1,2]. The detection of DNA hybridization via the differential pulse voltammetric (DPV) method has become very popular in the analytical field, because this method is able to detect genomic DNA and is easier as compared to the conventional polymerase chain reaction (PCR) DNA amplification technique [3,4,5]. The electrochemical method could lead to an amperometric response at low target concentration levels e.g., femtomolar or attomolar levels, with good precision, which is of great significance in DNA detection for clinical diagnosis of genetic diseases, infectious agents or forensic analysis [6].

In order to construct a high-performance electrochemical biosensor, the materials used to construct the matrices for biomolecule immobilization must possess characteristics like high electrical conductivity and compatibility with a variety of biological molecules [7,8,9]. Nano-sled materials including semiconductor quantum dot (QD) nanocrystals are among the most widely studied electron-transporting materials for their use to improve the electrochemical performance of the working electrode [10,11]. QD nanomaterials have been recently shown to enhance the electrochemical enzyme-based biosensor performance. Various QD nanoparticles, e.g., Mn^2+^ doped CdS, ZnS, CdTe-CdS core-shell QDs were found to have increased the electron transfer rate at the electrode surface without the use of electron mediator with >7-fold more sensitivity than gold nanoparticles (AuNPs)- and graphene nanosheets-based enzymatic biosensors. In addition, QDs-based immobilization matrices are able to provide a benign environment to retain the biomolecules’ activity and native conformation in the immobilized format [12,13,14,15].

The ultrafast electron transfer mechanism in QD nanoparticles has inspired researchers to employ QD semiconducting material in developing versatile ultrasensitive electrochemical DNA biosensors. A previous study reported the use of PbS QDs as DNA probe labels which were oxidized upon specific DNA hybridization events [16]. The amount of lead ion, which corresponds to the target DNA concentration was measured by an anodic stripping voltammetry method. Wang et al. [17], on the other hand, exploited three types of QDs nanoparticles, i.e., ZnS, CdS and PbS QDs, to simultaneously differentiate three different target DNAs via a stripping voltammetry electroanalytical technique, and was able to quantify target single stranded DNA (ssDNA) molecules at concentrations as low as 2 × 10^−13^ M. However, the construction of these biosensors involved the use of hazardous and toxic mercury that is not environmentally friendly.

Less toxic mercaptopropionic acid (MPA)-capped ZnS QDs was employed in this work as the immobilizing matrix for DNA probe to fabricate a voltammetric DNA biosensor. Anthraquinone monosulphonic acid (AQMS) redox intercalator was used as the DNA hybridization label. The screen-printed carbon paste electrode (SPE) was deposited with AuNPs in order to improve the SPE conductivity. Cysteamine was used as the organic binder to form covalent binding between immobilized AuNPs and carboxylated ZnS QDs, and facilitated by EDC carbodiimide coupling reagent. The self-assembled cysteamine that was bonded to the immobilized AuNPs also served to prevent non-specific absorptions of AQMS redox indicator on the electrode substrate. Detection of synthetic target DNA was performed via hybridization with the DNA probe, and was monitored based on AQMS reduction peak current response by using differential pulse voltammetry (DPV) method (Figure 1). The incorporation of MPA-ZnS QD nanoparticles in the proposed DNA biosensor design has significantly reduced the charge transfer resistance of the AuNPs modified electrode. The optimized electrochemical DNA biosensor has been applied for DNA sequence specific detection of male Arowana fish (*Scleropages formosus*) so that the gender of Arowana fish could be determined. In comparison with the proposed genetic gender identification of Arowana fish using electrochemical approach, the standard method, PCR, requires more skilled operators, more expensive reagents, and longer analysis time (up to 2.5 h). The arowana fish is a tropical freshwater fish from Southeast Asia that is believed to bring good luck and prosperity through positive feng shui energy because it resembles the traditional Chinese Dragon and that it has its own name among aquarium hobbyists–Dragonfish [18]. Its high economic value causes the increased demand in technology for distinguishing the gender as it is a crucial step for breeding, especially during the early stages of life. Thus, our research of using DNA-based rapid biosensors could significantly contribute to the development of the Arowana fish breeding industry. Indeed, there was a previously published report on similar research in determining the Arowana fish gender using carrageenan-polypyrrole-gold nanoparticles composite as the DNA immobilizant, whereby the synthesis of polypyrrole-based conducting polymer involved the use of toxic reagents and complicated reaction control that led to unwanted pollution to the environment. The interaction between biologicals and polypyrrole membrane could be modulated by a variety of factors including synthesis conditions and dopant choice, while thorough cleaning of the resulting membrane is required to eliminate toxic remnants, such as monomers or oligomers from the synthesis step prior to use [4].

## 2. Materials and Methods 

### 2.1. Reagents and Instrumentation

1-ethyl-3-(3-dimethylaminopropyl) carbodiimide (EDC), gold nanopowder (<100 nm particle size, 99.9%) and hydrochloric acid (HCl) were purchased from Sigma. Potassium dihydrogen phosphate (KH_2_PO_4_), dipotassium hydrogen phosphate (K_2_HPO_4_) and colloidal AuNPs were received from Merck. Sodium hydroxide (NaOH) and anthraquinone monosulphonic acid (AQMS) were obtained from Systerm and Acros, respectively. All the chemicals were of analytical reagent grade and used as received without further purification. All the aqueous solutions were prepared by using deionized water purified with Milli-Q water purification system. The 20-base pair ssDNA molecules were purchased from Sigma-Aldrich. The Arowana fish DNA samples were obtained from the Department of Fisheries and Freshwater Fisheries Research Center (PPPAT), Penang—Malaysia. The DNA sequences used in this work were similar to the previously reported study by Wang et al. [17]. These DNA oligonucleotide sequences are listed as follows:DNA probe: 5′ GGGGCAGAGCCTCACAACCT (AmC3)Target DNA: 5′ AGGTTGTGAGGCTCTGCCCC15% mismatched bases DNA: GAT TTG TGA GGC TCT GCC CC30% mismatched bases DNA: GAT TTG ACT GCC TCT GCC CC90% mismatched bases DNA: GAT TTG ACT GCC TCT CGT CCNon-complementary DNA: 5′ GGATGGACGAAGCGCTCAGGArowana fish DNA probe: 5’-TAA CTC AAAA GTA GAA TAG AAC A ATG [aminC3]

All the electrochemical measurements with DNA biosensor were performed on the AUTOLAB PG12 (AUT 71681) potentiostat/galvanostat. Ag/AgCl electrode saturated with 3 M KCl was used as the reference electrode, Pt electrode served as the auxiliary electrode and SPE modified with AuNPs, ZnS QDs, organic linkers and DNA probes was used as the working electrode. Cyclic voltammetry (CV) and electrochemical impedance spectroscopy (EIS) were used to investigate the effect of QDs nanoparticles, bifunctional ligands and DNAs on the conductivity of AuNPs-modified SPE. Optimization of the DNA biosensor based on mercaptopropionic acid (MPA)-capped ZnS QDs modified AuNPs (MPA-ZnS QDs/AuNPs) electrode was then carried out by means of differential pulse voltammetry (DPV).

### 2.2. Synthesis of ZnS Quantum Dot Nanocrystals

The water soluble ZnS QDs were prepared according to previously published protocols as reported by Eka et al. [19]. The MPA-functionalized ZnS QDs were synthesized by reacting the required amount of MPA with 100 mL of Zn(NO_3_)_2_ solution under vigorous stirring and nitrogen aeration for 20 min. The solution pH was maintained at pH 7.0 with 0.1 M NaOH. Then, some 100 mL of the Na_2_S solution was introduced dropwise into the solution under stirring and left to react overnight. The colloidal MPA-capped ZnS QDs obtained was then heated at 165 °C for 4 h and left to cool to ambient temperature (25 °C) before placing at 4 °C for long term storage.

### 2.3. Preparation of MPA-ZnS QDs/AuNPs-Based Electrochemical DNA Biosensor

The DNA biosensor was first prepared through layer-by-layer deposition of AuNPs. About 4 µL of AuNPs colloidal solution was drop-coated onto the SPE and air-dried at ambient conditions before coating with the next AuNPs layer. About ten layers of AuNPs were coated to form multilayer AuNPs assembly on the SPE surface. Then, the AuNPs-modified SPE (AuNPs-SPE) was immersed in 400 μL of 0.02 M cysteamine solution for 2 h to form the self-assembled chemisorbed monolayer of cysteamine on the AuNPs-SPE. Covalent binding of the MPA-ZnS QDs to the cysteamine-modified AuNPs-SPE was then achieved by soaking the electrode in a solution containing 300 μL of 0.1 mM MPA-ZnS QDs, 100 μL of 0.1M EDC and 100 μL of 0.1 M phosphate buffer solution (PBS) at pH 7, and incubating at 4 °C for 24 h in a refrigerator. The MPA-ZnS QDs-modified AuNPs-SPE (MPA-ZnS QDs/AuNPs-SPE) was later immobilized with aminated DNA probe by dipping the electrode into a solution containing 15 μL of 100 μM DNA probe, 100 μL of 0.1 M EDC and 185 μL of 0.01 M PBS (pH 7) for 24 h at 4 °C. DNA hybridization was carried out by reacting the MPA-ZnS QDs/AuNPs-based DNA biosensor with 300 μL of target DNA in 0.01 M PBS at pH 7 for 2 h followed by immersing the DNA biosensor in 0.01M PBS (pH 7) containing 1 mM AQMS redox intercalator for 1 h. Prior to DPV measurement, the DNA biosensor was washed with copious amounts of 0.01 M PBS (pH 7) to remove non-specifically adsorbed AQMS redox probe. DNA hybridization was investigated within the AQMS reduction potential range of −1.0 V to 0.0 V versus Ag/AgCl electrode in 0.01 M PBS (pH 7) containing 0.1 M KCl at the scan rate of 0.1 V s^−1^ by using DPV technique. The electrochemical behavior of each extended layer on the SPE was studied by using CV method at the scan rate range between 30 mV s^−1^ and 300 mV s^−^^1^ in 0.01 M PBS (pH 7) containing 1 mM AQMS and 0.1 M KCl.

### 2.4. Effect of AuNPs and MPA-ZnS QDs Loadings

Different amounts of AuNPs suspension in ethanol i.e., 0, 8, 16, 24, 32, 40, 48 and 56 μL (equivalent to 0, 0.03, 0.05, 0.08, 0.10, 0.13, 0.16 and 0.19 mg AuNPs) were deposited separately onto the SPE surface and air-dried at room temperature. The cyclic voltammogram of each AuNPs-SPE was then obtained in 0.1 M H_2_SO_4_ at the scan rate of 100 mV s^−1^. All these AuNPs-SPEs were then immersed separately in 0.01 M PBS (pH 7) containing 1 mM AQMS for 1 h before DPV measurement in 0.01 M PBS (pH 7) containing 0.1 M KCl at the scan rate of 100 mV s^−1^ and potential window between −1.0 V and 0.0 V. To optimize MPA-ZnS QDs loading on the AuNPs-SPE surface, the electrode was immersed in a series of 0.01 M PBS (pH 7) containing 0.1 M EDC with different MPA-ZnS QDs concentration from 0–0.06 μM. The DPV experiment was then conducted in 0.01 M PBS at pH 7 containing 1 mM AQMS and 0.1 M KCl at the scan rate of 100 mV s^−^^1^. The AuNPs-SPEs modified with different amounts of MPA-ZnS QDs were then immobilized with 5 μM of DNA probe followed by DNA hybridization with 5 µM of target DNA and immersed in 1 mM AQMS redox intercalator for 1 h. The DNA biosensor DPV response was performed in 0.01 M PBS (pH 7) containing 0.1 M KCl at the scan rate of 100 mV s^−1^. 

### 2.5. DNA Biosensor Response Time and Regeneration Study

The time it takes for DNA hybridization and AQMS intercalation will determine the response time of the DNA biosensor. DNA hybridization duration was determined by reacting the DNA biosensor with 5 µM target DNA for 30–150 min followed by soaking the DNA biosensor in 0.01 M PBS containing 1 mM AQMS at pH 7 for 1 h, whilst the AQMS intercalation duration was examined by dipping the DNA biosensor in 0.01 PBS (pH 7) containing 1 mM AQMS for between 15 min and 75 min after hybridization with target DNA. This was followed by washing with an abundant amount of 0.01 M PBS (pH 7) and the DPV response was conducted in 0.01 M PBS (pH 7) containing 0.1 M KCl at a scan rate of 100 mV s^−1^. Regeneration of the DNA biosensor was carried out by dissociating the immobilized dsDNA on the surface-modified SPE by using the temperature effect and NaOH regeneration solution. Three different temperatures at 63.3, 68.3 and 73.3 °C selected based on ±5 °C of the lower and upper DNA melting temperature (T_m_) and three different NaOH concentrations at 0.01, 0.001, and 0.0001 M were used to break the hydrogen bonding between complementary bases that hold the two DNA strands. The dsDNA-immobilized MPA-ZnS QDs/AuNPs-SPE was first immersed in the pre-heated water bath (30 s) or NaOH solution (1 min) for regeneration of the electrochemical DNA biosensor prior to soaking the DNA biosensor again in the 5 μM target DNA solution for 2 h followed by immersion in 1 mM AQMS solution for 1 h. The DPV response of the DNA biosensor was then measured in 0.01 M PBS containing 0.1 M KCl at pH 7 and scan rate of 100 mV s^−1^.

### 2.6. Optimization of Cysteamine and DNA Probe Concentrations

The DNA biosensors with different cysteamine loadings were prepared by immersing the AuNPs-SPE in 400 μL of cysteamine solution in the concentration rage of 0.005–0.05M for 2 h before further modifications with 0.1 mM MPA-ZnS QDs, 5 µM DNA probe, 0.1 M EDC, 5 µM target DNA and 1 mM AQMS. The DPV experiment was then performed in 0.01 M PBS containing 0.1 M KCl at pH 7 and scan rate of 100 mV s^−^^1^. The effect of DNA probe concentration on the DNA biosensor response was studied by immobilizing 0.5, 1, 2, 3, 4, 5 and 6 μM DNA probe separately on different MPA-ZnS QDs/AuNPs-SPEs in the presence of 0.1 M EDC at pH 7 and leaving for 24 h at 4 °C before they were used for the detection of 5 μM target DNA by using DPV technique.

### 2.7. pH and Buffer Capacity Effects on the DNA Biosensor Response

The pH of the DNA hybridization medium and electrolyte buffer was optimized between pH 5 and pH 10 by using 0.01 PBS to yield the maximum DNA biosensor response. The PBS concentration at pH 7 was then varied between 0.5 mM and 50 mM to prepare the DNA hybridization and electrolyte buffers and tested with the DNA biosensor. The DNA biosensor DPV response was taken at the potential scan rate of 100 mV s^−1^ in the presence of 0.1 M KCl after reaction with 5 µM target DNA and 1 mM AQMS.

### 2.8. Biosensor Lifetime and Selectivity Study

For biosensor shelf life estimation, about 21 DNA biosensors were prepared under the same conditions with 5 µM of immobilized DNA probe and stored in a refrigerator at 4 °C. Three DNA biosensors were tested with 1 × 10^−6^ µM of target DNA on the first day of every week, and the DPV response of each DNA biosensor was recorded at 100 mV s^−1^ potential scan rate until a decline in DNA biosensor response was observed. The DNA biosensor selectivity was assessed by reacting with 5 µM target DNA,15% mismatched bases DNA, 30% mismatched bases DNA, 90% mismatched bases DNA and non-complementary DNA separately for 3 h followed by soaking in 1 mM AQMS for 1 h. The DPV response of the DNA biosensor was measured in 0.01 M PBS (pH 7) containing 0.1 M KCl at the scan rate of 100 mV s^−1^. Finally, the calibration range of the electrochemical DNA biosensor was developed by exposing the DNA biosensor to a series of target DNA concentrations from 1 × 10^−10^ to 1 × 10^−1^ μM under the optimum conditions.

### 2.9. Determination of Arowana Fish Gender

Then, an electrochemical Arowana fish DNA biosensor was fabricated based on the optimum conditions of the electrochemical synthetic DNA biosensor fabrication. The male Arowana fish probe DNA was immobilized on the surface of the modified SPE/nAu/cys/ZnS QDs and hybridized using the similar method as presented previously. A calibration curve for the detection of Arowana fish genders was determined in a range of Arowana fish DNA target concentrations from 1 × 10^−18^ to 1 × 10^−2^ μM. The Arowana DNA biosensor was used to determine the gender of Arowana fish by immersing the electrodes into sample solutions which had been treated with a similar condition of target for the calibration curve. The samples of Arowana fish DNA were diluted using 300 µL of 10 mM PBS pH 7 and sonicated to dissociate the double strand DNA. After the hybridization process, the electrodes were immersed in 1 mM AQMS solution for 1 hour and the response was detected using the DPV method in a mixed solution of 0.01 M PBS pH 7 and KCl 0.1 M.

## 3. Results and Discussions

### 3.1. Electrode Optimization

The electrochemical behaviour of each extended layer on the SPE was evaluated using CV method by applying various scan rates of 30 mV s^−^^1^ to 300 mV s^−^^1^. Figure 2a represents the scan rate dependent AQMS cathodic peak current (i_pc_) at each extended layer on the SPE surface. The reduction current of anthraquinone redox indicator was proportional to the scan rate for every substance layer deposited onto the SPE. The linear behavior of the AQMS i_pc_ signal with scan rate indicates that the system was controlled by a diffusion process [20]. The quantity of current flow on the modified SPE surface can be estimated from the slope of the plot based on the Randles–Sevcik equation. The electron transfer rate profile of the bare SPE and modified SPEs is displayed in Figure 2b. 

As can be seen, the bare electrode has the lowest slope value of 0.076 µAV^−^^1^ s, and the electron transfer rate improved significantly at 0.301 µAV^−^^1^ s when the electrode surface was covered with multilayer AuNPs. The presence of AuNPs on the SPE surface has improved the electrode conductivity, thereby enabling efficient transport of electrons on the SPE surface [21]. When the non-conductive cysteamine was deposited on top of the AuNPs-SPE, it exerted electrical resistance onto the electrode, leading to a substantial decline in the electron transfer rate at 0.224 µAV^−^^1^ s. The increase in the electron transfer rate can be clearly seen when the AuNPs-SPE was further modified with a layer of semiconducting MPA-ZnS QDs. The rate of electron transport at 0.338 µAV^−^^1^ s of the MPA-ZnS QDs/AuNPs-SPE was slightly higher than the AuNPs-SPE, proving the ability of the MPA-ZnS QDs to enhance the conductivity of the surface-modified SPE. The rate of electron transport remained stable after the MPA-ZnS QDs/AuNPs-SPE was immobilized with aminated ssDNA probes or double-stranded DNAs (dsDNAs) due to electrical conductivity conferred by DNA molecule itself, whereby the stacked base pairs system channelled the electron transfer to the electrode surface [22]. Buk et al. argued that the nano-scaled particles are able to reduce the distance between redox site of a biomolecule and the electrode, and the rate of electron transfer is inversely dependent on the exponential distance between them [23].

The DPV response of the SPE after every modification with AuNPs, cysteamine, MPA-ZnS QDs, DNA probe, target DNA and non-complementary DNA and 1 h AQMS intercalation duration is shown in Figure 2c. No noticeable AQMS reduction signals were noted for AuNPs-SPE and MPA-ZnS QDs/AuNPs-SPE as they possessed negative charges from the respective carboxylate and carboxyl groups that provided electrostatic repulsion with AQMS redox intercalator of the same charge and they repelled each other from interaction. Therefore, the effect of the non-specific adsorption of AQMS intercalator on the modified electrode surface could be neglected.

The reduction peak of AQMS increased sharply after the dsDNA-immobilized MPA-ZnS QDs/AuNPs-SPE was exposed to 1 mM of AQMS solution for 1 h. The strong and sharp AQMS reduction peak indicates the successful intercalation of AQMS redox probe into the immobilized dsDNA on the MPA-ZnS QDs/AuNPs-SPE [24]. Negligible AQMS DPV response was obtained for DNA probe-immobilized MPA-ZnS QDs/AuNPs-SPE suggesting no interaction between DNA probe and the AQMS redox probe. Non-complementary DNA did not show significant DPV response as the non-matching sequence has less or no hybridization during the recognition event.

### 3.2. Optimization of AuNPs and MPA-ZnS QDs Loadings

By increasing the AuNP loading on the SPE surface from 0.0–0.19 mg, the AuNPs i_pc_ signal at approximately +0.9 V increased proportionally (Figure 3a) as more and more conductive pathways were created for electron transfer through the immobilized AuNPs. However, increasing the AuNPs loading from 0.03–0.10 mg on the SPE showed a decreasing AQMS reduction signal at ~−0.6 V (Figure 3b) as both AuNPs and AQMS are negatively charged, resulting in a force that caused them to repel each other [25]. This could reduce the amount of non-specific AQMS binding on the SPE. The carbon-based bare SPE appeared to absorb anionic AQMS indicator strongly due to the positively charged cathode surface, as the opposite charges attracted one another. In view of the minimal non-specific AQMS binding on the AuNPs-SPE with 0.13 mg immobilized AuNPs, it was employed for further DNA biosensor fabrication. 

Optimization of the immobilized MPA-ZnS QDs quantity on the SPE was evaluated based on the function of MPA-ZnS QDs as the immobilization matrix and electron transfer media. As Figure 3c indicates, both MPA-ZnS QDs/AuNPs-SPE and DNA biosensor showed increasing signal trends with increasing MPA-ZnS QDs loading, and the DPV current signal started to decline slightly or become leveled off when the available amine functional groups from the cysteamine-modified AuNPs-SPE had entirely reacted with the carboxyl group of the MPA-functionalized ZnS QDs. About 0.03 µM MPA-ZnS QDs was found to be optimum for the construction of the DNA biosensor.

### 3.3. DNA Biosensor Response Time and Rehybridization of DNA Biosensor

The response time for DNA hybridization and AQMS intercalation are presented in Figure 4. The DNA biosensor response gradually increased from 30 min to 90 min and reached a saturation response between 90 min and 120 min due to the completed DNA hybridization reaction on the electrode surface. However, when the DNA hybridization duration was prolonged to 150 min, the DNA biosensor response decreased. This might be due to the over loading of the dsDNA on the electrode surface, which may have developed an electron transport barrier. The time required for optimum AQMS intercalation into the immobilized dsDNA was found to be 45 min as a steady state response was attained from 45 min onwards of the AQMS immersion time.

The average rehybridization percentage of the DNA biosensor following the heating treatment was below 60% (Figure 4c). The complete rehybridization could not be achieved due to the denaturation of the DNA molecules at hot temperature. The destruction of the DNA biosensor might also be due to the release of thiol functional group from the cysteamine moiety attached to the AuNPs surface [26]. Higher rehybridization percentages of the DNA biosensor could be achieved by using diluted NaOH concentrations (0.001 and 0.0001 M) as the DNA biosensor regeneration solution compared to the high concentration of NaOH at 0.01 M, where most of the immobilized dsDNAs were permanently denatured in the high pH condition, meaning that regeneration of the DNA biosensor was not possible (Figure 4d).

### 3.4. Effect of Cysteamine and DNA Probe Loadings on the DNA Biosensor Response

The self-assembled monolayer of cysteamine on the AuNPs via thiol functional group has been widely studied to allow further modification of the electrode surface to fabricate highly sensitive electrochemical sensors/biosensors for trace analysis [26,27]. Figure 5a shows the effect of cysteamine concentration on the DNA biosensor based on AQMS reduction current signal. 

As cysteamine is the bifunctional ligand required to covalently bind to the MPA-ZnS QDs for subsequent DNA probe binding, the increasing amounts of cysteamine loaded to the AuNPs electrode promoted higher MPA-ZnS QDs and DNA probe loadings to the electrode for hybridization with greater amounts of target DNAs. However, when an amount of cysteamine at >0.04 M was loaded, it resulted in the decline of DNA biosensor response which may be ascribed to the action of insulators, i.e., the immobilized cysteamine, which has increased the resistance to the current flow through the electrode. Hence, optimum cysteamine loading at 0.04 M was used to modify the AnNPs-SPE before the DNA probe immobilization step. The effect of DNA probe loading on the electrochemical DNA biosensor response is illustrated in Figure 5b. At small DNA probe loading (0.5 µM), the low DPV response obtained implies a low DNA hybridization reaction rate at the electrode surface. By increasing the DNA probe loading from 1.0 µM to 6.0 µM, the DNA biosensor response increased drastically, and the current response tends to further increase with a higher DNA probe loading. This could be attributed to the high surface area of the immobilized MPA-ZnS QDs that provide large amounts of active sites for covalent coupling with a large amount of DNA probes.

### 3.5. pH and Buffer Concentration Effects

Effects of pH, buffer capacity and ionic strength of the DNA hybridization and electrolyte buffers were optimized in order to obtain the highest possible electrochemical DNA biosensor response. The electrochemical DNA biosensor response was found to be optimum in neutral PBS solution (Figure 6a). Both acidic and alkaline buffers were not favorable for DNA hybridization and AQMS intercalation reactions due to the irreversible denaturation of DNA structure in both acidic and basic media. The DNA biosensor response was then optimized in terms of buffer concentration by using PBS at pH 7. Figure 6b shows that the DNA biosensor response increased as the PBS concentration increased from 0.5 mM to 10 mM. This was due to the fact that high buffer capacity can promote rapid DNA hybridization by reducing the repulsive force between the negatively charged sugar-phosphate backbone of target DNA and the DNA probe [28]. However, the extremely high PBS concentration at 50 mM created a high ionic strength environment which was unfavorable for the optimum DNA hybridization reaction and subsequent AQMS intercalation reaction to take place.

### 3.6. Long Term Stability and Selectivity of the DNA Biosensor Based on MPA-ZnS QDs/AuNPs-SPE

A study of long-term stability was carried out in order to quantify the storage stability of the DNA electrode. Figure 7a represents the DPV current response of the DNA biosensor towards the detection of 1 × 10^−^^6^ µM target DNA for seven weeks under optimal conditions. The DNA biosensor gave the highest current response on the first operational day and the DNA biosensor DPV response started to reduce slightly in the second week, whereby about 86.8% of the DNA biosensor initial response was still achievable. A further decrease in the DPV peak current of the DNA biosensor was still perceived in the third storage week of the DNA electrode, and the DNA biosensor remained constant at ~75.5% until the seventh storage week of the DNA biosensor. 

Selectivity of the electrochemical DNA biosensor based on MPA-ZnS QDs/AuNPs-SPE was evaluated with non-complementary DNA and mismatched DNA strands, and compared with the DNA biosensor signal with target DNA. The highest current response was obtained with target DNA as the fully matched sequences allowed maximum AQMS intercalation into the immobilized DNA duplexes (Figure 7b). The DNA biosensor response as indicated by AQMS i_pc_ signal at ~−0.6 V declined with the increasing percentage of mismatched bases in DNAs, i.e., from 15% base mutation in oligonucleotide and onwards, due to the lesser amount of electroactive AQMS intercalated into dsDNA structure, with the mismatched sequences becoming less effective in hybridizing with their complementary DNA probe immobilized on the QDs-based electrode. The DNA biosensor with non-complementary sequences generated negligible response as did the immobilized DNA probe alone.

### 3.7. Dynamic Linear Response Range of the DNA Biosensor

The optimized synthetic DNA probe-immobilized MPA-ZnS QDs/AuNPs-SPE was used to generate a calibration curve for the target DNA. The DNA biosensor showed increasing DPV response with target DNA concentration from 1 × 10^−10^ µM to 1 × 10^−3^ µM, and the DNA biosensor response saturated thereafter with higher target DNA concentrations (Figure 8). A linear response range of the DNA biosensor was observed between 1 × 10^−9^ µM and 1 × 10^−3^ µM target DNA (R^2^ = 0.9848) with a limit of detection (LOD) estimated at 1 × 10^−^^11^ µM target DNA. The sensitivity of the biosensor can be determined based on the slope value of the calibration curve, i.e., 1.2885 ± 0.12 µA/decade (µM). The average reproducibility relative standard deviation (RSD) of each calibration point of the electrochemical DNA biosensor using a new electrode for each DNA concentration testing was calculated at 7.4%.

### 3.8. Arowana Fish Gender Determination

In this study, a calibration curve for the determination of Arowana fish gender was measured based on a 26 base sequence, aminated DNA probe of Arowana. The sensitivity of the biosensor was determined to be 1.125 µA (R^2^ = 0.9785) at response linear 1 × 10^−^^11^ to 1 × 10^−^^6^ µM. The profile of biosensor response at different target concentrations of Arowana fish and the plot of current versus concentration of target are presented in Figure 9 and Figure 10.

Based on this calibration curve, the LOD of the biosensor was determined to be about 1 × 10^−^^11^ µM. Both electrochemical DNA biosensors were constructed using a similar system and measured at similar optimum conditions. Nevertheless, they had different LOD, sensitivity and width of dynamic range. The different values might be due to the difference in length of DNA sequence used for the immobilization. The length of DNA probe sequence seemed to affect the sensitivity, width of dynamic range and LOD.

The electrochemical DNA biosensor which was immobilized with male Arowana fish DNA probe sequence was used to determine the gender of the fish. The result of the gender determination was based on the LOD value. A response current obtained above LOD value is considered to be that of a male Arowana fish.

### 3.9. Comparison with Other Electrochemical DNA Biosensors Based on QD Nanomaterials

Most electrodes fabricated based on self-assembled thiolated compounds on the AuNPs are sustained from pivotal bottleneck, whereby the resistance of the electrodes increased as a result of surface coverage by adsorbed ligand layers that are much less conductive. Based on Table 1, previously reported electrochemical DNA biosensors based on QD nanomaterials utilized QDs as electrochemical response indicators/labels, for instance, ZnS, CdS and PbS QDs. The unique design of the proposed DNA biosensor was the incorporation of MPA-ZnS QDs semiconducting material for reduction of resistance of the AuNPs modified electrode (Cys-AuNPs-SPE), which may be conferred by the AuNPs surface ligand i.e., the insulating Cys moieties. The proposed DNA detection strategy had clearly improved the performance of the DNA biosensor because it had broadened the dynamic range and lowered the detection limit compared to other previously developed electrochemical DNA biosensors by almost a thousand-fold (Table 2). The proposed MPA-ZnS QDs modified AuNPs electrode would serve as a promising platform with great potential for investigation with some other biological or chemical materials to form a biological or chemical active receptor phase to produce versatile electrochemical nanosensors.

## 4. Conclusions

The modification of AuNPs-SPE with MPA-ZnS QDs has demonstrated enhancement of the electron transfer rate due to the high electrical conductivity of MPA-ZnS QD semiconducting nanomaterials particles. The inclusion of MPA-ZnS QDs nanomaterial assembled on the electrode surface reduced the charge transfer resistance apart from acting as a high specific surface area platform for high loading of immobilized DNA molecules. This has enabled improvement in the overall DNA biosensor performance. A linear proportionality between cathodic peak current of AQMS and target DNA concentration was successfully established by using the proposed MPA-ZnS QDs/Cys/AuNPs-SPE electrode with a wider linear response range and lower detection limit. The electrochemical DNA biosensor can be used for low level detection of DNA at an attomolar detection limit and has been successfully used to determine Arowana fish gender.

## Figures and Tables

**Figure 1 biosensors-12-00650-f001:**
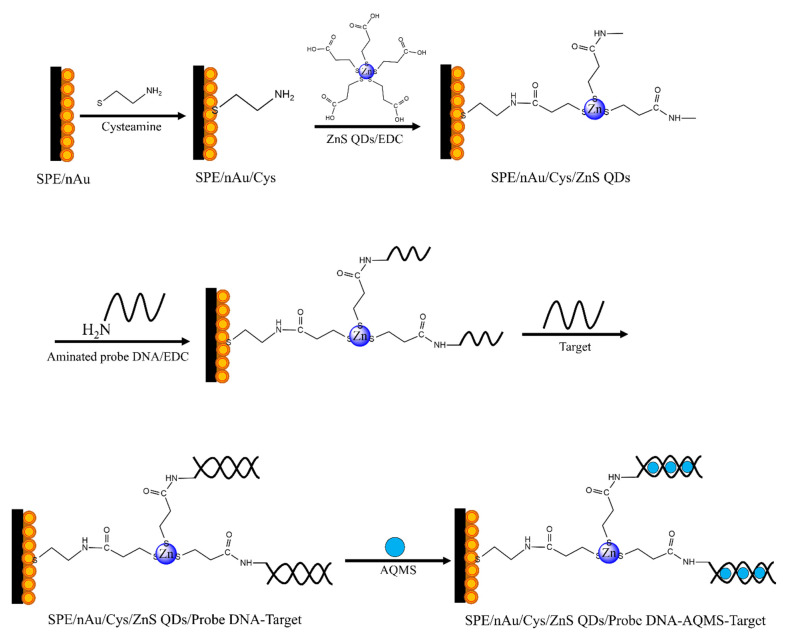
The stepwise construction of the electrochemical DNA biosensor based on multilayer AuNPs, cysteamine linkers and MPA-capped ZnS QDs semiconducting nanoparticles.

**Figure 2 biosensors-12-00650-f002:**
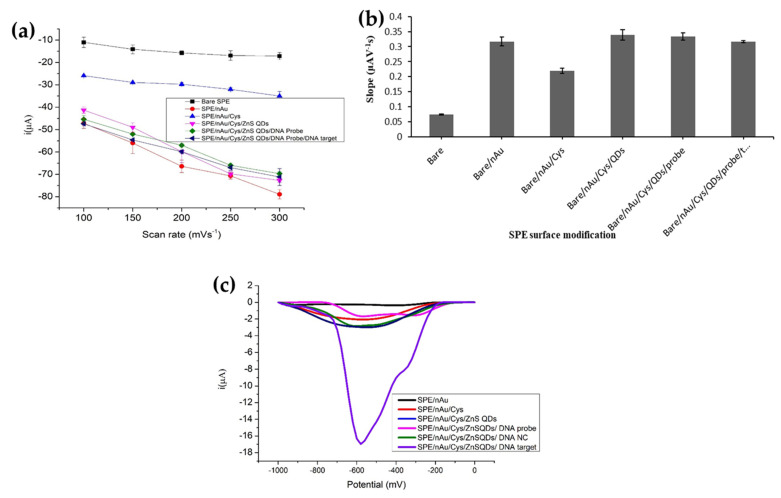
(**a**) The electron transfer rates estimated from the Randles–Sevcik equation for bare SPE and SPEs modified with AuNPs, cysteamine linkers, ZnS QDs semiconducting nanoparticles and DNA molecules by using 1 mM AQMS redox indicator. (**b**) The sensitivity dependent AQMS i_pc_ response at each extended layer on the SPE at scan rate range between 30 mV s^−^^1^ and 300 mV s^−^^1^ in 0.01 M PBS at pH 7 containing 1 mM AQMS and 0.1 M KCl. (**c**) The DPV response of SPE after every modification with AuNPs, cysteamine, MPA-ZnS QDs, DNA probe, target DNA and non-complementary DNA and 1 h AQMS intercalation duration. The DPV measurement was conducted in 0.01 M PBS solution containing 0.1 M KCl at a scan rate of 100 mV s^−^^1^.

**Figure 3 biosensors-12-00650-f003:**
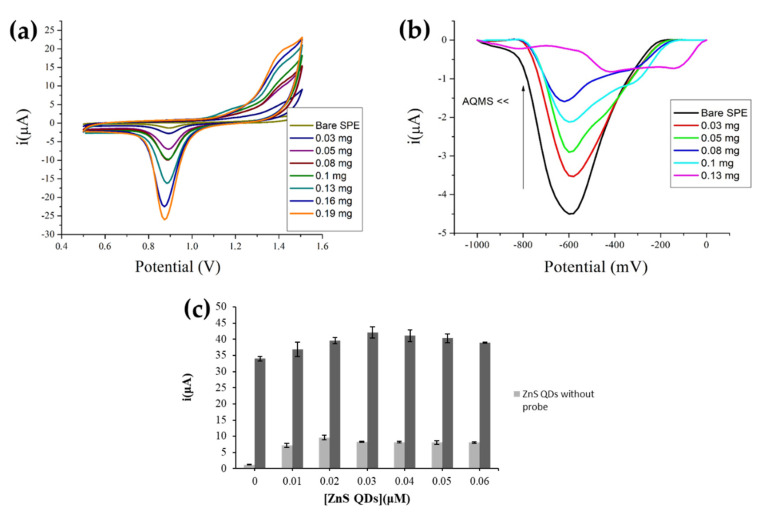
(**a**) The cyclic voltammograms of AuNPs-SPEs with different amounts of immobilized AuNPs. The CV measurement was done in 0.1 M H_2_SO_4_ at a scan rate of 100 mV s^−1^. (**b**) The differential pulse voltammograms of AuNPs-SPEs with various AuNPs loadings after exposure to 1 mM AQMS for 1 h. The DPV measurement was conducted in 0.01 M PBS (pH 7) containing 0.1 M KCl at a scan rate of 100 mV s^−1^. (**c**) The DPV responses of MPA-ZnS QDs/AuNPs-SPE and DNA biosensor at different immobilized MPA-ZnS QDs amounts. The DPV measurements were carried out in 0.01 M PBS (pH 7) containing 0.1 M KCl and 1 mM AQMS for MPA-ZnS QDs/AuNPs-SPE and 0.01 M PBS (pH 7) containing 0.1 M KCl for DNA biosensor at the scan rate of 100 mV s^−1^.

**Figure 4 biosensors-12-00650-f004:**
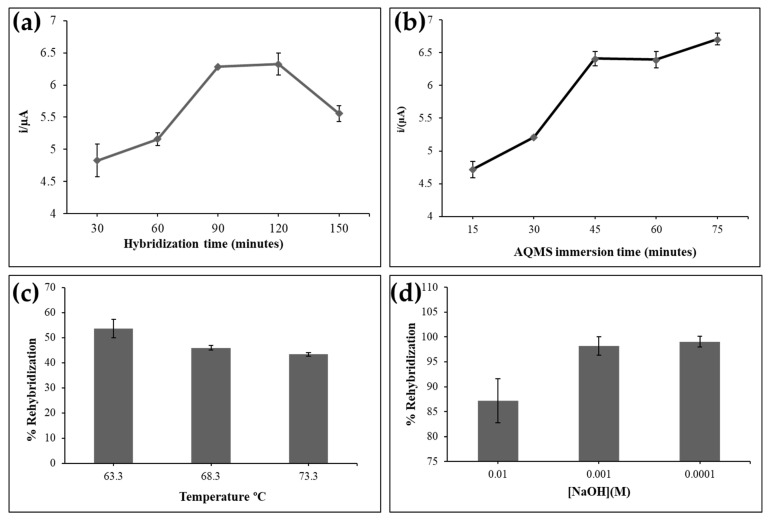
(**a**) The DNA biosensor response trend for a DNA hybridization period of 30–150 min by using 5 µM DNA probe and 5 µM target DNA. (**b**) The DNA biosensor response after exposure to 1 mM AQMS label from 15–75 min in 0.01 M PBS (pH 7). The DPV measurement was conducted in 0.01 M PBS (pH 7) containing 0.1 M KCl at the scan rate of 100 mV s^−1^. The rehybridization profiles of the electrochemical DNA biosensor by using (**c**) temperature effect at 63.3, 68.3 and 78.3 °C and (**d**) NaOH regeneration solution at 0.01 M, 0.001 M and 0.0001 M.

**Figure 5 biosensors-12-00650-f005:**
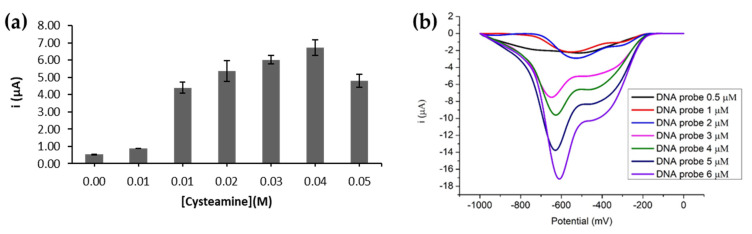
(**a**) The DPV response trend of electrochemical DNA biosensor incorporated with various concentrations of cyteamine from 0.00-0.05 M. The DPV experiment was performed in 0.01 M PBS (pH 7) containing 0.1 M KCl at the scan rate of 100 mV s^−^^1^. (**b**) Effect of different DNA probe concentrations from 0.5 µM to 6.0 µM on the detection of 5 µM target DNA. The DPV scanning was conducted in 0.01 M PBS containing 0.1 M KCl at pH 7 and scan rate of 100 mV s^−^^1^ after the DNA biosensor was immersed in 1 mM AQMS for 1 h.

**Figure 6 biosensors-12-00650-f006:**
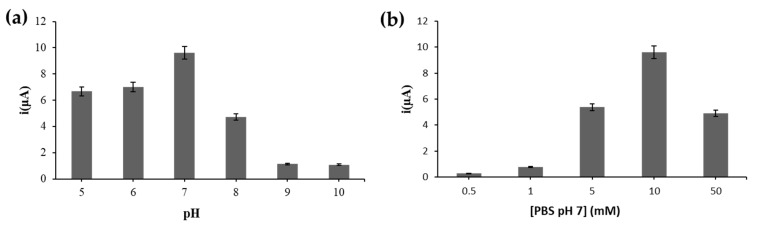
(**a**) Effects of pH from pH 5 to pH 10 and (**b**) PBS concentration from 0.5 mM to 50 mM containing 0.1 M KCl on the DNA biosensor response. The DPV response was recorded using a scan rate of 100 mV s^−^^1^.

**Figure 7 biosensors-12-00650-f007:**
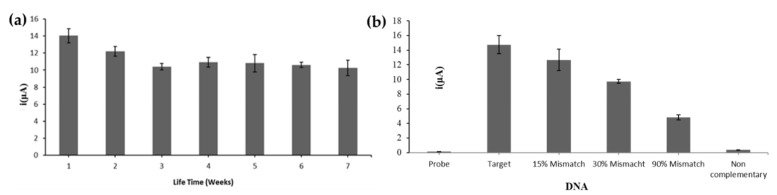
(**a**) The DPV current response of the DNA electrode towards the detection of 1 × 10^−6^ µM target DNA for seven weeks under the optimal conditions. (**b**) The DNA biosensor response towards 5 µM target DNA, mismatch DNA and non-complementary DNA with 3 h DNA hybridization time and 1 h AQMS intercalation duration. The DPV measurement was conducted in 0.01 M PBS (pH 7) containing 0.1 M KCl at the scan rate of 100 mV s^−1^.

**Figure 8 biosensors-12-00650-f008:**
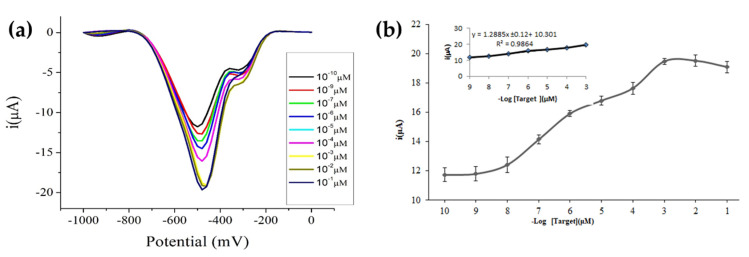
(**a**) DPV response of the electrochemical DNA biosensor at different target DNA concentrations. (**b**) The response curve of the electrochemical DNA biosensor established with different concentrations of target DNA from 1 × 10^−10^ µM to 1 × 10^−1^ µM. The inset shows the linear calibration curve of the DNA biosensor from 1 × 10^−9^ µM to 1 × 10^−3^ µM target DNA.

**Figure 9 biosensors-12-00650-f009:**
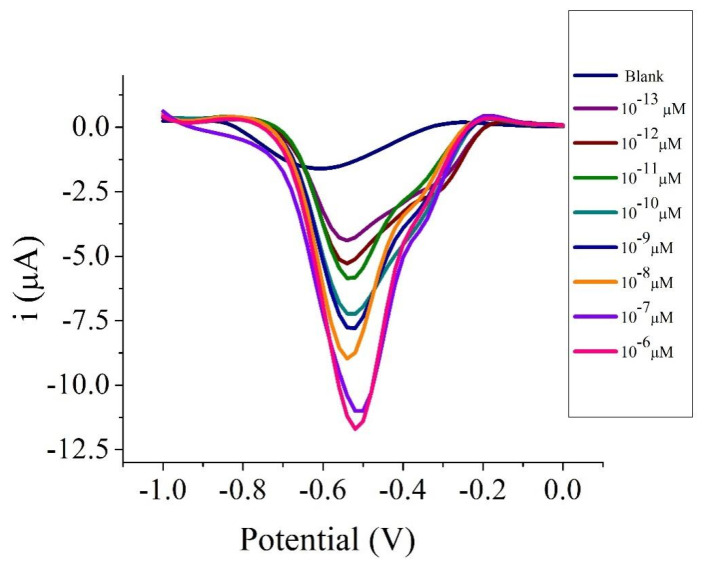
DPV of electrochemical DNA biosensor toward Arowana DNA target concentrations of 1 × 10^−^^13^ to 1 × 10^−^^6^ µM. The biosensors were scanned in 10 mM PBS at pH 7 and 0.1 M KCl using a scan rate of 0.1 V s^−^^1^.

**Figure 10 biosensors-12-00650-f010:**
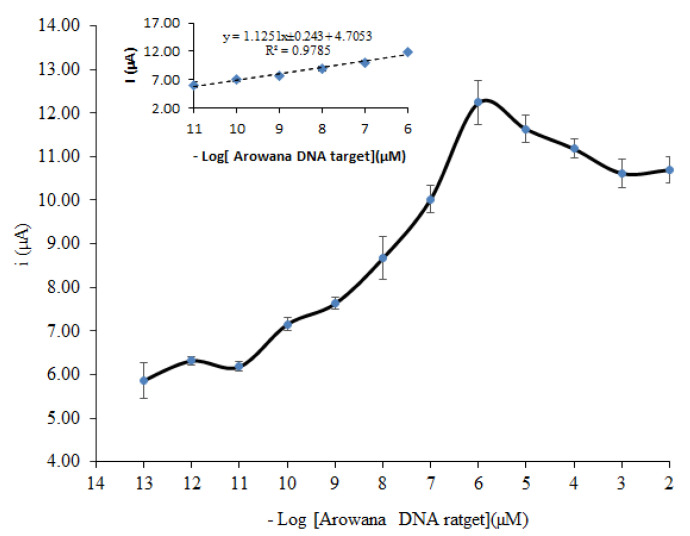
Calibration curve of electrochemical DNA biosensor using 5 µM targets hybridized with different concentrations of target from 1 × 10^−^^13^ µM to 1 × 10^−^^2^ µM. The biosensors were run in 10 mM PBS at pH 7 and 0.1 M KCl using a scan rate of 0.1 V s^−^^1^.

**Table 1 biosensors-12-00650-t001:** Determination of Arowana fish gender using electrochemical DNA biosensor based on self-assembled cysteamine on gold nanoparticles and covalent bonding of ZnS QDs.

Sample’s Number	Response Current (µA)	% RSD n = 3	Gender Estimation
519	8.64	7.14	Male
521	8.47	3.15	Male
523	8.48	5.17	Male
526	7.44	1.68	Male

**Table 2 biosensors-12-00650-t002:** The comparison of electrochemical DNA biosensor performance with the previously reported electrochemical DNA biosensors based on QD nanomaterials.

DNA Immobilization Matrices	Labels	Linear Range (M)	Detection Limit (M)	Ref.
DNA detection using CdS nanocluster as labeling tag	CdS	2.25 × 10^−^^13^–2.25 × 10^−^^7^	2 × 10^−^^12^	[29]
DNA detection using PbS nanocluster as nano particle tag	PbS	2.25 × 10^−^^12^–2.25 × 10^−^^9^	3 × 10^−^^13^	[30]
Using in situ plated mercury-coated GCE	ZnS, CdS and PbS QDs	1 × 10^−^^15^–1 × 10^−^^12^	4 × 10^−^^11^	[17]
The use of carbon nanotubes as modifier on the transducer surface	CdS	8 × 10^−^^12^–4 × 10^−^^9^	2.75 × 10^−^^12^	[31]
The self-assembly of MPA on gold electrode	PbS	1.2 × 10^−^^11^–4.8 × 10^−^^8^	4.38 × 10^−^^12^	[32]
chitosan-entrapped SPE modified with Au nanoparticles	CdSe	5 × 10^−^^12^–5 × 10^−^^7^	6.5 × 10^−^^13^	[33]
MPA-ZnS QDs/Cys/AuNPs-SPE	AQMS	1 × 10^−^^15^–1 × 10^−^^9^	1 × 10^−^^17^	Present work

## Data Availability

Underlying data can be requested to the first author (E.S.).

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
