# Peer review of "Electrochemical DNA Biosensor Based on Mercaptopropionic Acid-Capped ZnS Quantum Dots for Determination of the Gender of Arowana Fish"

_biosensors, 2022, doi:10.3390/bios12080650_

Round 1

Reviewer 1 Report

In this paper, a new electrochemical DNA biosensor was reported via the self-assemble of mercaptopropionic acid capped ZnS quantum dot. The MPA-ZnS QDs modified with AuNPs-SPE was identified to have enhanced the electron transfer rate due to the high electrical conductivity of MPA-ZnS QD semiconducting nanomaterials particles. In this research, different influencing factors were researched to get the optimized sensing conditions. And the material was used for the arowana fish gender determination. This research is adequate and interesting, but the following questions should be paid attention before publication.

1.      The author mentioned that similar research has been reported for Arowana fish gender determination with another nanoparticle composite. So, the authors should highlight their own advantages compared with the reported nanoparticle.

2.      I am wondering how the authors characterize to make sure they got the target materials. Detailed introduction should be given.

3.      There are some typos and mistakes in this manuscript. For example, Fig 2b should be Fig 2a in Line 261, Fig 2c should be Fig 2b in Line 268. The b) c) in Figure 2. The unit of scan rate in Line 346. The legend for the black bar in Fig 3c is missing. Please check carefully to avoid this kind of mistakes.

4.      The biosensor was used for determining the genders of fish. Why only the male samples were shown in Table 1? How the female samples are shown in the response current should also be given as a contrast.

5.      The author mentioned that two biosensors might have different LOD, sensitivity and width of dynamic range in Line 485. Is this kind of difference manageable and controllable?

6.      The figures with high resolution should be provided for publication.

Author Response

Dear Reviewer,

We thank you for the suggestions and comments which have helped us improving the scientific value of our manuscript. Kindly find the details attached.

Reviewer 2 Report

This work includes the mien of Due to the high electrical conductivity of MPA-ZnS QDs, modifying AuNPs-SPE with them increases electron transfer rate. MPA-ZnS QDs nanomaterial formed on the electrode surface lowered charge transfer resistance and provided a high-specific surface-area platform for DNA immobilization. Overall, DNA biosensor performance has improved. Using the suggested MPA-ZnS QDs/Cys/AuNPs-SPE electrode, a linear relationship between AQMS cathodic peak current and target DNA concentration was obtained. The electrochemical DNA biosensor has been utilized to identify Arowana fish gender at attomolar detection levels. It can be accepted after lifting the following significant concerns:

a.       The title should be clarified with a specific meaning that correctly represents the whole context.

b.      An abstract must be enriched via further valuable quantitative highlights, which pave the way for persuading the audiences to read the full text. Moreover, the abstract should determine this study's differences from similar reports.

c.       Keywords should not include the title's contents.

Introduction:

d.      Introduction should be started differently to attract a wide range of audiences. Four recommendations, which could be considered the model, is Nano-Micro Letters volume 13, 18 (2021) https://link.springer.com/article/10.1007/s40820-020-00533-y.

Methodology:

GOOD

Results:

e.       Raw data for review should be presented.

Discussion:

f.        Outlook and future perspectives are not up to date.

References:

Give more up-to-date references with the exclusion of possible self-citations.

Tables / Figures:

g.      I prefer having figures at the end not within the context for the revised version.

h.      Repetitions for obtaining the bar charts should be mentioned.

i.        The captions are desired to be further enriched via info, making figures independently understandable instead of relying on the context.

Author Response

Dear Reviewer 2,

We thank you for the constructive comments. We have incorporated your comments into our revised manuscript (as highlighted in red or track change feature). Also, we have provided our responses in details attached.

Reviewer 3 Report

This work introduced an electrochemical method to detect the gender of Arowana Fish which is of highly economical value. Below are a couple of comments:

1. The method proposed is for gender detection of Arowana Fish, the authors are suggested to introduce the traditional or standard method for the gender detection of Arowana Fish, and compare with it.

2. There are some small typos in the text: line 43-44: 'alsoindependent' should be 'also independent'; line 45: 'thedetection' should be 'the detection'; line 298: 'thatthe' should be 'that the'.

Author Response

Thank you for taking the time of evaluating our manuscript. Please find our responses as an attached file.

Round 2

Reviewer 1 Report

The authors answered my questions properly. And the paper looks good from my viewpoint. 

Author Response

Thank you for the final evaluation of our manuscript. And thank you for the positive feedback pertaining to our revised manuscript. But since there is no more additional comments requiring the revision of our manuscript, we did not make any modification to the current manuscript, except for the title revision as requested by the academic editor. The title has been changed to:

Electrochemical DNA Biosensor based on Mercaptopropionic Acid-Capped ZnS Quantum Dots for Determination of the Gender of Arowana Fish